# Towards Task-Consistent Open-Vocabulary Adaptation in Video Recognition

## Abstract

Transferring CLIP for open-vocabulary video recognition has shown impressive effectiveness. To fit the video domain, the model undergoes fine-tuning on a video dataset and is expected to generalize well on data with unseen categories. However, this fine-tuning paradigm overlooks the variations of the representation space beyond the training distribution, leading to the sub-optimal adaptation effect. In this paper, we introduce *TACO*, a simple yet effective framework to mitigate the potential negative effects induced by the inconsistency between fine-tuning and evaluation objectives. We formulate a more concrete adaptation principle by delving into the deficiencies of existing paradigms. Specifically, we propose a task decoupling method that mitigates the knowledge overfitting by incorporating a specialization projection. Moreover, we offer new insights into the preservation of the generalization and technically introduce *Relative Structure Distillation*, which maintains the consistent relative structure between in-distribution and out-of-distribution representation spaces through knowledge distillation. Our proposed *TACO* establishes state-of-the-art performance on diverse benchmarks under cross-dataset and base-to-novel settings. Code will be released.

## 1 Introduction

Open-vocabulary learning aims to identify novel visual concepts defined by language vocabularies unseen during the training phase. This setting has been driven by the rapid progress of large-scale Vision–Language Models (e.g., CLIP (Radford et al., 2021), ALIGN (Jia et al., 2021)), which learn aligned multimodal representations by pretraining on massive image-text pairs. With its strong generalization to unknown concepts, such foundation models offer great practicality for real-world applications and have been widely adapted to various downstream tasks (Zhou et al., 2022; Gu et al., 2022; Cho et al., 2024; Lin et al., 2024). Inspired by this progress, recent studies have explored adapting CLIP for general video recognition (Ni et al., 2022; Rasheed et al., 2023; Zhu et al., 2024), avoiding the costly computation and video data collection for pretraining from scratch.

For open-vocabulary video recognition, while CLIP provides a decent starting point for context understanding, effective adaptation remains challenging, specifically in empowering models to accurately capture the temporal dynamics encoded in videos of unknown categories. To achieve this, it is believed that the key lies in *introducing video-specific knowledge while preserving the original pretrained generalization*. Following this principle, recent studies (e.g., Open-VCLIP (Weng et al., 2023), FROSTER (Huang et al., 2024)) typically seek a middle ground between specialization and generalization, thereby alleviating the catastrophic forgetting problem induced by fine-tuning (Kumar et al., 2022; Zheng et al., 2023) and yielding promising results. However, although underpinning many recent advances, the essence of this principle remains insufficiently clarified, providing little concrete direction for method evolution. As a natural extension, our questions are: *(i) Can fine-tuning learn new knowledge suitable for open-vocabulary video recognition? (ii) What is the essence of preserving the pretrained generalization during fine-tuning?*

We explore the above questions by rethinking the inconsistency between standard fine-tuning and the open-vocabulary task, where models are tuned to pursue decent performance on training data but evaluated on out-of-distribution (OOD) categories. Since the training dataset is limited in scale and diversity, such inconsistency problem is inherently present in existing fine-tune paradigms and leads to sub-optimal open-vocabulary adaptation effects. By observing performance changes when mixing

Joint Representation Space

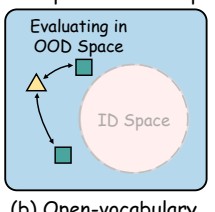 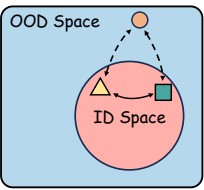

(a) Standard Fine-tuning  (b) Open-vocabulary Evaluation  (c) Ours Fine-tuning

Figure 1: A representation space illustration of the inconsistency between the standard fine-tuning and open-vocabulary evaluation objectives. Triangles indicate the visual embedding, squares represent the text embedding, and circles denote sampled OOD embeddings.

the evaluation vocabulary, it reveals that the knowledge introduced by standard fine-tuning exhibits severe in-distribution (ID) bias. We then delve into the impact of feature space selection and show that keeping alignment with the initial OOD semantic space is crucial for preserving generalization.

Building upon these findings, we present *TACO*, a simple yet effective framework towards task-consistent open-vocabulary video adaptation. Our inspiration stems from observing the deficiencies of existing fine-tuning methods relative to a hypothesized ideal adaptation framework, where the video and text data used for fine-tuning are virtually unlimited. By remedying these deficiencies, *TACO* makes refinements in both introducing new knowledge and preserving generalization. Specifically, instead of directly mapping video inputs to the optimization space, we propose decoupling the adaptation task into learning generalized knowledge and optimizing specific objectives. This is achieved by incorporating a *Specialization Projection* on top of the visual encoder during fine-tuning, separating the encoder's output from the optimization space and thereby mitigating the in-distribution bias. We discuss specific projection designs in the context of VLMs adaptation. As for the generalization capability, we offer new insights into the role of fine-tuning the text encoder and the essence of preserving generalization. Technically, we propose a novel distillation objective termed *Relative Structure Distillation*. Distinct from existing distillation terms, the proposed objective regularizes the relative structure between ID and constructed OOD spaces to be consistent with its initial state, closing the gap between training and evaluation objectives. Overall, our proposed *TACO* is highly concise and exhibits great scalability. It can be seamlessly integrated into existing video learners across various architectures and model sizes, delivering significant improvements.

Our main contributions can be summarized as follows:

- We introduce *TACO*, a simple yet effective framework for open-vocabulary video adaptation. *TACO* aims to mitigate the potential negative effects induced by the inconsistency between fine-tuning and evaluation objectives, which has been overlooked in previous studies.
- We revisit the existing fine-tuning paradigm, identify its deficiencies in terms of introducing new knowledge and preserving generalization, and formulate a more explicit adaptation principle (§ 2).
- We propose a space decoupling method that mitigates the knowledge overfitting by incorporating Specialization Projection. Moreover, we present Relative Structure Distillation, which effectively preserves generalization by keeping the relative alignment of OOD space during fine-tuning (§ 3).
- Our proposed *TACO* sets a new state-of-the-art performance on cross-dataset and base-to-novel settings across multiple benchmarks. Comprehensive ablation studies demonstrate the effectiveness and versatility of our method. (§ 4)

## 2 ANALYSIS: REVISITING OPEN-VOCABULARY ADAPTATION FOR VIDEO

### 2.1 PRELIMINARY: ADAPTING CLIP FOR VIDEO RECOGNITION

To transfer the powerful generalization of the CLIP model (Radford et al., 2021) to the video domain, recent studies have fine-tuned its joint embedding space using video-text data and obtained promising results (Ni et al., 2022; Rasheed et al., 2023; Zhu et al., 2024). We briefly describe the standard fine-tuning paradigm in this context. Consider a CLIP-based video learner composed of a visual encoder $f_{\theta_v}(\cdot)$ and a text encoder $f_{\theta_t}(\cdot)$, where $\theta_v$ and $\theta_t$ are the parameters of each encoder.

Given a video $V \in \mathbb{R}^{T \times H \times W \times 3}$ with $T$ frames and the corresponding text description $C \in \mathcal{S}_{ft}$ embedded in a set of pre-defined templates (e.g., *"This is a video about []"*), we extract the visual embedding $\boldsymbol{v} \in \mathbb{R}^D$, the text embedding $\boldsymbol{c} \in \mathbb{R}^D$, and compute their similarity $\text{sim}(\boldsymbol{v}, \boldsymbol{c})$ as follows:

$$\boldsymbol{v} = f_{\theta_v}(V), \quad \boldsymbol{c} = f_{\theta_t}(C), \quad \text{sim}(\boldsymbol{v}, \boldsymbol{c}) = \frac{\langle \boldsymbol{v}, \boldsymbol{c} \rangle}{\|\boldsymbol{v}\| \|\boldsymbol{c}\|}, \tag{1}$$

where $D$ is the dimension of the joint embedding space. During fine-tuning, the video-specific knowledge is injected by maximizing the similarity if $V$ and $C$ are matched, otherwise minimizing it. Formally, the training objective can be written as:

$$\mathcal{L}_{CE} = \mathbb{E}_{(V,C) \sim \mathcal{D}}[\, CE(\, \sigma(\text{sim}(\boldsymbol{v}, \boldsymbol{c})), \, one\_hot(C)\,)\,], \tag{2}$$

where $\mathcal{D}$ is the fine-tuning dataset, $CE(\cdot, \cdot)$ is the cross-entropy loss, and $\sigma$ is the softmax operation.

## 2.2 INCONSISTENCY BETWEEN OPEN-VOCABULARY ADAPTATION AND EVALUATION

Following the above fine-tuning paradigm, great efforts have been made to adapt CLIP for open-vocabulary video recognition (Weng et al., 2023; Huang et al., 2024; Zhu et al., 2024; Yu et al., 2025). These methods typically conduct fine-tuning on the Kinetics-400 (Kay et al., 2017) dataset, and the derived model is expected to generalize well on test data with unseen categories $C_{test} \in \mathcal{S}_{test}$, where $\mathcal{S}_{ft} \cap \mathcal{S}_{test} = \emptyset$. However, the standard fine-tuning objective is optimized restrictively within the in-distribution (ID) space, neglecting the basic setting of evaluating open-vocabulary tasks in the OOD space. We argue that such inconsistency between training and evaluation objectives leads to suboptimal effects in both introducing new knowledge and preserving generalization.

**In-distribution bias in learning new knowledge.** To endow the model with generic video knowledge for temporal modeling, the fine-tuning data is expected to be as large-scale and diverse as possible. In existing works, while Kinetics-400 (K400) can serve as a good proxy on the video side, its text space is bounded by a fixed number of 400 action categories. During fine-tuning, the optimization objective forces the visual encoder to learn a mapping function from the video input space to a narrow embedding space. As a result, the new knowledge learned by the visual encoder severely overfits the limited ID text space and is not suitable for open-vocabulary tasks. To demonstrate this, we measure how ID and OOD performance change as we expand the vocabulary used for evaluation. As shown in Figure 2, when expanding the corresponding vocabulary of each dataset with

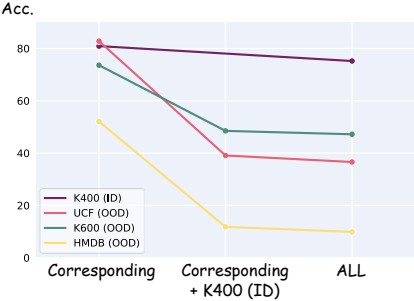

Figure 2: Accuracy drops when expanding the vocabulary used for evaluation.

ID categories (i.e. K400), OOD performance significantly drops while ID performance is marginally affected. This phenomenon suggests that the visual embedding of the adapted model suffers from severe in-distribution bias.

**A closer look at preserving generalization.** To better understand the essence of preserving generalization, we investigate the impact of each fine-tuned encoder on generalization by replacing it with the corresponding CLIP's original encoders during evaluation. We show the harmonic mean of zero-shot performances in Figure 3 (a). Interestingly, replacing the fine-tuned text encoder with CLIP's text encoder has almost no impact on the results (66.3% vs. 66.2%), which suggests that the OOD generalization of the adapted model is grounded in CLIP's

|  | FT Text | CLIP Text |  |  | FT Text | CLIP Text |
|---|---|---|---|---|---|---|
| FT Visual | 66.3 | 66.2 | | FT Visual | 68.4 | 67.3 |
| CLIP Visual | 61.4 | 60.9 | | CLIP Visual | 62.8 | 60.9 |
| (a) Standard FT model | | | | (b) Our FT model | | |

Figure 3: Replacing the encoders of the standard fine-tuning model and our model with the original CLIP encoders.

original semantic space. However, fine-tuning inevitably leads to changes in the OOD representation space (evidenced in Figure 4), as the representations of the test data are not orthogonal to the ID subspace spanned by the training data (Kumar et al., 2022). This change may further distort the alignment between visual and text embeddings in OOD space and result in diminished generalization. Based on these findings, we hypothesize that *maintaining the representation alignment in OOD space is crucial for preserving generalization.* As shown in Figure 3 (b), properly regularizing the OOD space with our method can further enhance the generalization, validating our hypothesis.

**The role of fine-tuning the text encoder.** Our hypothesis naturally raises a question: Since the generalization relies on the original CLIP text space, can we freeze the text encoder to prevent the alignment shift in OOD space? Unfortunately, the answer is no, as previous works show that fine-tuning the text encoder can bring improvements to zero-shot performance (Rasheed et al., 2023; Huang et al., 2024). To prove this, we visualize the similarity distributions between visual embeddings from the CLIP model and various fine-tuned models in Figure 4, and calculate the overall similarities between text embeddings. It can be observed that freezing the text encoder during fine-tuning causes the visual embeddings to deviate substantially from those of CLIP, whereas fine-tuning the text

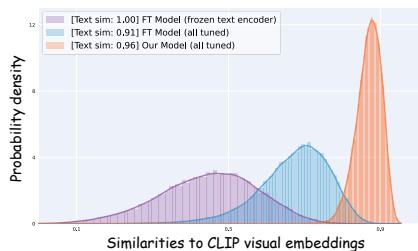

Figure 4: Similarity distributions of visual embeddings between the CLIP model and various fine-tuned models in OOD space (UCF, HMDB, and K600).

encoder can effectively mitigate such deviation. Based on this, we argue that the role of fine-tuning the text encoder is to *relieve the overfitting in visual embeddings, rather than learning new textual knowledge*. However, there is still a noticeable deviation for both fine-tuned visual and text embeddings in OOD space, which degrades the generalization capability. We believe such deviation should be further curbed during fine-tuning, which was overlooked in previous studies.

## 3 METHODOLOGY

Building upon the above findings, we conclude that an effective open-vocabulary adaptation strategy requires considering: *(i) Introducing new knowledge while mitigating its overfitting to known categories. (ii) Maintaining the representation alignment in OOD space for preserving generalization.* With this in mind, we propose a simple yet effective framework *TACO*, as presented in Figure 5. Details will be elaborated in the following sections.

### 3.1 SPECIALIZATION PROJECTION FOR MITIGATING KNOWLEDGE OVERFITTING.

In the standard fine-tuning paradigm, the representation space is typically identical to the optimization space, as the output visual embeddings are directly fed into the objective function for parameter optimization. This paradigm is desirable when the optimization space is capable of delivering extensive supervision to the visual representations. Conversely, the narrow supervision from limited text categories will collapse the open distribution of CLIP's visual representations into a restricted space, leading to overfitting.

To address this, our insight is to decouple the representation space from the optimization space by incorporating a *Specialization Projection* between the two spaces. The representation space is expected to learn more generic knowledge around the original CLIP space, while the optimization space is dedicated to minimizing the fine-tuning objective. During evaluation, we discard the projection head for more generalized representations. Specifically, we first encode the video input $V$ into the representation space with the visual encoder $f_{\theta_v}(\cdot)$, and then transform it to the optimization space with a projection network $h(\cdot)$:

$$\hat{\boldsymbol{v}} = h(f_{\theta_v}(V)) + f_{\theta_v}(V), \tag{3}$$

where $\hat{\boldsymbol{v}} \in \mathbb{R}^D$ is the projected visual embedding. Our design is inspired by self-supervised contrastive learning (Chen et al., 2020a;b; Oquab et al., 2023), where an MLP is placed on top of the encoder during the pre-training stage, and discarded afterwards when fine-tuning on downstream tasks. However, applying such a strategy for VLMs adaptation is non-trivial since there is only one training phase. In this context, we need to consider the effective decoupling effect while keeping the alignment between text and pre-projection visual embeddings. To keep the alignment, we apply the residual connection (He et al., 2016) and remove the non-linear activation in the projection head. Different from the critical role of the non-linear activation in contrastive learning, we find that it distorts the representation alignment when the text encoder is tunable. The projection head is parameterized as two consecutive linear layers. We apply different learning rates to the projection head and visual encoder to achieve more effective decoupling. Besides, we do not incorporate projection to the text encoder, since its role is not to learn new textual knowledge as described in Section 2.2.

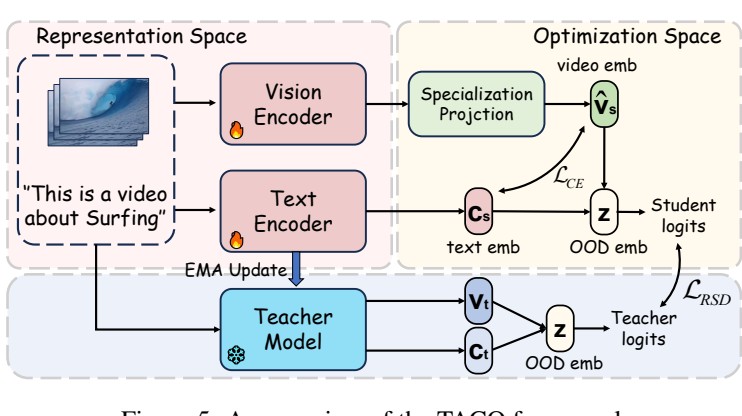

Figure 5: An overview of the TACO framework.

## 3.2 RELATIVE STRUCTURE DISTILLATION FOR PRESERVING GENERALIZATION

Existing fine-tuning paradigm naturally partition the semantic space into ID and OOD spaces, where a model fine-tuned on the known training distribution is expected to generalize to unseen categories. While achieving great advances, these studies overlook the importance of OOD space for preserving generalization. Based on our findings, our insight is that the fine-tuning process should regularize the entire semantic space to protect the representation alignment. To achieve this, we propose *Relative Structure Distillation*, a novel regularizer that maintains the consistent relative structure over the entire semantic space during fine-tuning by knowledge distillation (Hinton et al., 2015).

**Relative structure distillation.**  To preserve the pre-trained generalization during adaptation, one direct approach is to constrain the adapted model from deviating too far from the pre-trained version by mimicking its logits or features (Huang et al., 2024; Addepalli et al., 2024). For example, one can adopt the frozen CLIP as the teacher model and extract the normalized visual embedding $v_t$ and text embedding $c_t$. The Kullback-Leibler (KL) divergence loss can be utilized to match the distribution between teacher and student logits:

$$\mathcal{L}_{KL} = D_{KL}(\,\sigma(v_t\,c_t^\top/\tau)\,||\,\sigma(v_s\,c_s^\top/\tau)\,), \tag{4}$$

where $\sigma$ denotes the softmax operation, $\tau$ is a temperature parameter, $v_s$ and $c_s$ are the normalized embeddings from the adapted model. However, existing distillation methods typically focus on the matching within the training distribution since the distillation data is identical to the training data, leaving them ineffective in suppressing deviations beyond the training distribution. To address this, we propose anchoring the relative structure of the ID and OOD spaces with generated OOD semantic points. Formally, given a video and text input, we first obtain the normalized teacher embeddings $v_t, c_t \in \mathbb{R}^D$ and student embeddings $\hat{v}_s, c_s \in \mathbb{R}^D$, respectively. Note that $\hat{v}_s$ is derived from the projected visual embedding in Equation 3. Subsequently, we sample $N$ normalized OOD semantic embeddings across the entire semantic space, denoted as $z \in \mathbb{R}^{N \times D}$. The generated OOD points do not need to be associated with the training data, but expected to cover enough semantics. We take $z$ as anchors and regularize its relative relations to the ID embeddings with a modified KL objective:

$$\mathcal{L}_{RSD} = D_{KL}(\,\sigma(v_t\,\boxed{z^\top z}\,c_t^\top/\tau)\,||\,\sigma(\hat{v}_s\,\boxed{z^\top z}\,c_s^\top/\tau)\,). \tag{5}$$

Instead of distilling the embedding similarities within the training distribution, our objective aims to keep the consistency of relative structure between ID and OOD spaces throughout the fine-tuning.

**Generation of OOD Embeddings.**  To cover enough semantics, the OOD embeddings should ideally span the entire semantic space. One possible way is to sample from a large authentic corpus. For example, leveraging the textual descriptions in existing large-scale video-text datasets (e.g. We-bVid (Bain et al., 2021)) or prompting large-language models (Hurst et al., 2024) to generate potential video categories. But the sampled texts may not uniformly span the semantic space and require an additional embedding process. To address this, we formulate the generation of OOD semantic embeddings as a problem of uniform sampling on a unit hypersphere. Specifically, leveraging the rotational invariance of the Gaussian distribution, we draw $N$ samples from the standard Gaussian

Table 1: Performance comparison with state-of-the-art methods under the base-to-novel setting. "HM" indicates the harmonic mean of the accuracy on base and novel sets.

| Method | Venue | K400 | | | HMDB | | | UCF | | | SSv2 | | |
|---|---|---|---|---|---|---|---|---|---|---|---|---|---|
| | | BASE | Novel | HM | BASE | Novel | HM | BASE | Novel | HM | BASE | Novel | HM |
| CLIP (Radford et al., 2021) | ICML'21 | 62.3 | 53.4 | 57.5 | 53.3 | 46.8 | 49.8 | 78.5 | 63.6 | 70.3 | 4.9 | 5.3 | 5.1 |
| ActionCLIP (Wang et al., 2021) | arXiv'21 | 61.0 | 46.2 | 52.6 | 69.1 | 37.3 | 48.5 | 90.1 | 58.1 | 70.7 | 13.3 | 10.1 | 11.5 |
| X-CLIP (Ni et al., 2022) | ECCV'22 | 74.1 | 56.4 | 64.0 | 69.4 | 45.5 | 55.0 | 89.9 | 58.9 | 71.2 | 8.5 | 6.6 | 7.4 |
| VPT (Ju et al., 2022b) | ECCV'22 | 69.7 | 37.6 | 48.8 | 46.2 | 16.0 | 23.8 | 90.5 | 40.4 | 55.8 | 8.3 | 5.3 | 6.4 |
| AIM (Yang et al., 2022) | ICLR'23 | 74.6 | 62.5 | 68.0 | 64.0 | 51.6 | 57.1 | 89.8 | 76.4 | 82.6 | 8.5 | 7.9 | 8.2 |
| ST-Adapter (Pan et al., 2022) | NeurIPS'22 | 73.6 | 62.0 | 67.3 | 65.3 | 48.9 | 55.9 | 85.5 | 76.8 | 80.9 | 9.3 | 8.4 | 8.8 |
| ViFi-CLIP (Rasheed et al., 2023) | CVPR'23 | 76.4 | 61.1 | 67.9 | 73.8 | 53.3 | 61.9 | 92.9 | 67.7 | 78.3 | 16.2 | 12.1 | 13.9 |
| Open-VCLIP (Weng et al., 2023) | ICML'23 | 76.5 | 62.6 | 68.9 | 70.3 | 50.4 | 58.7 | 94.8 | 77.5 | 85.3 | 16.0 | 11.0 | 13.0 |
| FROSTER (Huang et al., 2024) | ICLR'24 | 77.8 | 64.3 | 70.4 | 74.1 | 58.0 | 65.1 | 95.3 | 80.0 | 87.0 | 18.3 | 12.2 | 14.6 |
| Open-MeDe (Yu et al., 2025) | ICCV'25 | 77.2 | 63.8 | 69.9 | 73.6 | 56.4 | 63.9 | 94.9 | 78.5 | 85.9 | 17.1 | 12.3 | 14.3 |
| **TACO (ours)** | | **78.2** | 63.8 | 70.3 | **74.4** | **62.3** | **67.8** | **95.8** | **82.2** | **88.5** | 17.7 | **12.4** | **14.6** |

distribution $\mathcal{N}(0, I)$ in each iteration and project onto the unit hypersphere with normalization. The sampled vectors can be regarded as generic OOD embeddings and directly applied to Equation 5.

**Design of the teacher model.** Previous methods suggest that using the frozen CLIP as the teacher model is a good choice for preserving generalization (Huang et al., 2024; Zheng et al., 2023). However, the distillation supervision from the frozen CLIP may hinder the model's ability in learning new knowledge. In our work, the teacher model is designed as an exponential moving average (EMA) model with its parameters updated at each iteration as $\tilde{\theta} = m\tilde{\theta} + (1 - m)\theta$, where $m$ is a momentum coefficient, $\tilde{\theta}$ and $\theta$ are the parameters of the teacher and student models, respectively. This mechanism ensures the teacher model remains consistent while evolving with new knowledge. The overall learning objective can be written as

$$\mathcal{L} = \mathcal{L}_{CE} + \lambda \mathcal{L}_{RSD}, \tag{6}$$

where we set $\lambda = 0.4$ by default.

## 4 EXPERIMENTS

### 4.1 EXPERIMENTAL SETUP

**Evaluation Protocols.** We thoroughly evaluate our method with two common protocols: *Cross-dataset* and *Base-to-novel* evaluation. *(i) Cross-dataset:* In this setup, the model is trained on Kinetics-400 (Kay et al., 2017) and then evaluated on other datasets with out-of-distribution vocabulary, including UCF-101 (Soomro et al., 2012), HMDB-51 (Kuehne et al., 2011), and Kinetics-600 (Carreira et al., 2018). For UCF and HMDB, we evaluate on both the full dataset and three validation splits. For K600, we adopt the three splits provided by (Chen & Huang, 2021). Each split contains 160 categories sampled from 220 unseen categories. *(ii) Base-to-novel:* In this protocol, a dataset is divided into two disjoint category sets: base classes and novel classes. The model is tuned on base classes and evaluated on both base and novel classes. Evaluation datasets including K400, UCF, HMDB, and Something-Something v2 (Goyal et al., 2017). During inference, we sample 3 temporal clips with a center crop (i.e. $3 \times 1$ views) per video.

**Implementation Details.** We use the CLIP (Radford et al., 2021) pre-trained ViT-B/16 and ViT-L/14 models in our experiments. During fine-tuning, we sparsely sample 8 frames as the video input. The pre-processing includes random cropping and resizing to the size of $224 \times 224$, along with random horizontal flips and random grayscale. We adopt AdamW (Loshchilov & Hutter, 2017) as the optimizer with a weight decay of 0.2. The initial learning rate is set to $3.75 \times 10^{-6}$ with a total batch size of 192, following a half-period cosine learning rate decay. For parameters in the specialization projection, the initial learning rate is increased to $5 \times 10^{-5}$. Furthermore, we set the number of OOD semantic embeddings $N$ to 200 and the momentum coefficient $m$ to 0.9998. *Please see supplementary for more details.*

### 4.2 MAIN RESULTS

**Base-to-novel video recognition.** In Table 1, we compare our method with the state-of-the-art results under the base-to-novel setting, which reflects the model's joint ability to fit video-specific

Table 2: Zero-shot classification performance compared with the state-of-the-art methods under the cross-dataset setting, evaluated on the *validation splits* of UCF-101, HMDB-51, and Kinetics-600.

| Method | Venue | Encoder | Frames | UCF-101 | HMDB-51 | Kinetics-600 |
|--------|-------|---------|--------|---------|---------|--------------|
| ActionCLIP (Wang et al., 2021) | arXiv'21 | ViT-B/16 | 32 | 58.3±3.4 | 40.8±5.4 | 67.7±1.1 |
| A5 (Ju et al., 2022a) | ECCV'22 | ViT-B/16 | 32 | 69.3±4.2 | 44.3±2.2 | - |
| X-CLIP (Ni et al., 2022) | ECCV'22 | ViT-B/16 | 32 | 72.0±2.3 | 44.6±5.2 | 65.2±0.4 |
| ST-Adapter (Pan et al., 2022) | NeurIPS'22 | ViT-B/16 | 8 | 76.9±0.8 | 51.5±0.6 | 60.2±1.8 |
| Vita-CLIP (Wasim et al., 2023) | CVPR'23 | ViT-B/16 | 8/32 | 75.0±0.6 | 48.6±0.6 | 67.4±0.5 |
| ViFi-CLIP (Rasheed et al., 2023) | CVPR'23 | ViT-B/16 | 32 | 76.8±0.7 | 51.3±0.6 | 71.2±1.0 |
| OTI (Zhu et al., 2023) | ACMMM'23 | ViT-B/16 | 8 | 83.3±0.3 | 54.2±1.3 | 66.9±1.0 |
| Open-VCLIP (Weng et al., 2023) | ICML'23 | ViT-B/16 | 8 | 83.4±1.2 | 53.9±1.2 | 73.0±0.8 |
| MAXI (Lin et al., 2023) | ICCV'23 | ViT-B/16 | 16/32 | 78.2±0.8 | 52.3±0.7 | 71.5±0.8 |
| FROSTER (Huang et al., 2024) | ICLR'24 | ViT-B/16 | 8 | 84.8±1.1 | 54.8±1.3 | 74.8±0.9 |
| OST (Chen et al., 2024) | CVPR'24 | ViT-B/16 | 8 | 77.9±1.3 | 54.9±1.1 | 73.9±0.8 |
| MoTE (Huang et al., 2024) | NeurIPS'24 | ViT-B/16 | 8 | 83.4±0.7 | 55.8±0.9 | 70.2±0.6 |
| Open-MeDe (Yu et al., 2025) | ICCV'25 | ViT-B/16 | 8 | 83.7±1.3 | 54.6±1.1 | 73.7±0.9 |
| **TACO (ours)** | | ViT-B/16 | 8 | **85.6**±1.2 | **60.0**±0.5 | **77.0**±0.9 |
| X-Florence (Ni et al., 2022) | ECCV'22 | Florence | 32 | 73.2±4.2 | 48.4±4.9 | 68.8±0.9 |
| Text4Vis (Wu et al., 2023a) | AAAI'23 | ViT-L/14 | 8 | 82.6±0.7 | 52.4±0.4 | 72.1±0.9 |
| OTI (Zhu et al., 2023) | ACMMM'23 | ViT-L/14 | 8 | 88.1±1.0 | 59.3±1.7 | 70.6±0.5 |
| Open-VCLIP (Weng et al., 2023) | ICML'23 | ViT-L/14 | 8 | 87.6±1.2 | 59.0±0.6 | 81.1±0.8 |
| DiST (Qing et al., 2023) | ICCV'23 | ViT-L/14 | 32 | 74.9±0.8 | 57.5±1.6 | 75.0±0.7 |
| MoTE (Huang et al., 2024) | NeurIPS'24 | ViT-L/14 | 8 | 88.7±0.6 | 61.4±1.3 | 78.4±0.9 |
| **TACO (ours)** | | ViT-L/14 | 8 | **91.4**±0.7 | **64.2**±0.8 | **83.9**±0.7 |

Table 3: Zero-shot performance of UCF and HMDB on the *full* dataset. * indicates evaluation with the full validation set on HMDB.

| Method | Encoder | UCF | HMDB |
|--------|---------|-----|------|
| CLIP (Radford et al., 2021) | ViT-B/16 | 74.9 | 46.7 |
| AIM (Yang et al., 2022) | ViT-B/16 | 79.0 | 49.5 |
| ST-Adapter (Pan et al., 2022) | ViT-B/16 | 77.9 | 50.3 |
| Open-VCLIP* (Weng et al., 2023) | ViT-B/16 | 83.5 | 53.2 |
| FROSTER* (Huang et al., 2024) | ViT-B/16 | 85.0 | 54.5 |
| **TACO (ours)** | ViT-B/16 | **85.9** | **54.6** |
| Text4Vis (Wu et al., 2023a) | ViT-L/14 | 79.6 | 49.8 |
| BIKE (Wu et al., 2023a) | ViT-L/14 | 80.8 | 52.8 |
| OTI (Zhu et al., 2023) | ViT-L/14 | 88.3 | 55.8 |
| MoTE (Zhu et al., 2024) | ViT-L/14 | 89.4 | 56.3 |
| **TACO (ours)** | ViT-L/14 | **91.6** | **59.9** |

Table 4: Integrating our TACO with various adaptation methods can significantly improve the zero-shot generalization.

| Type | Method | UCF | HMDB | K600 |
|------|--------|-----|------|------|
| Adapter-based | ST-Adapter (Pan et al., 2022) | 77.9 | 50.3 | 60.2 |
| | + *TACO(ours)* | **80.5** | **53.1** | **73.9** |
| | Δ | +2.6 | +2.8 | +13.7 |
| Prompt-based | Vita-CLIP (Wasim et al., 2023) | 78.6 | 50.5 | 67.4 |
| | + *TACO(ours)* | **80.3** | **52.6** | **73.2** |
| | Δ | +1.7 | +2.1 | +5.8 |
| LoRA-based | CLIP + LoRA (Hu et al., 2022) | 80.5 | 50.4 | 71.2 |
| | + *TACO(ours)* | **83.1** | **54.4** | **74.7** |
| | Δ | +2.6 | +4.0 | +3.5 |
| Fully-tuned | Fine-tuned CLIP | 82.8 | 52.0 | 73.4 |
| | + *TACO(ours)* | **84.8** | **54.2** | **75.6** |
| | Δ | +2.0 | +2.2 | +2.2 |

biases while being adapted to unknown categories. Our method exhibits excellent performance on the K400, HMDB, and UCF datasets, primarily focusing on the improvements for novel categories. This demonstrates its ability to rapidly acquire generic video knowledge with a few less relevant samples. Besides, the temporal-heavy nature of SSv2 necessitates additional techniques (e.g., cross-frame attention (Weng et al., 2023; Huang et al., 2024)) to capture the fine-grained temporal dynamics. Since our method is built upon the original CLIP architecture, the performance of SSv2 does not show a notable improvement over other models. Overall, our method presents superior performance in the base-to-novel setting.

**Cross-dataset video recognition.** Table 2 presents comparisons with the state-of-the-art methods under the cross-dataset setting, which assesses the model's generalization towards out-of-distribution categories. Our method sets a new state-of-the-art performance across all datasets, yielding significant improvements over existing approaches. The excellent performance can be scaled up with the network architecture, indicating the effectiveness and scalability of our method. The same trend can also be observed when evaluating with the full dataset, as shown in Table 3. Overall, our adapted model exhibits remarkable OOD generalization, which can be attributed to the well-preserved structure of the representation space during the fine-tuning. As shown in Figure 4, our method can effectively mitigate the embedding deviations in OOD space and therefore improve the generalization capability.

Table 5: Ablation studies on the components and key details. We report the cross-dataset performance on UCF, HMDB, and K600 split1, using the ViT-B/16 network. $\text{UCF}_f$ and $\text{K600}_f$ denote the results of freezing the text encoder during the fine-tuning. Default settings are colored in gray.

(a) Effects of the proposed components.

| Spec. Proj. | $\mathcal{L}_{\text{RSD}}$ | UCF | HMDB | K600 |
|---|---|---|---|---|
| | | 82.8 | 52.0 | 73.4 |
| ✓ | | 83.7 | 53.2 | 74.8 |
| | ✓ | 84.5 | 52.9 | 75.2 |
| ✓ | ✓ | **84.8** | **54.2** | **75.6** |

(b) Various types of the Specialization Projection.

| Trpe | UCF | K600 | $\text{UCF}_f$ | $\text{K600}_f$ |
|---|---|---|---|---|
| None | 82.8 | 73.4 | 81.5 | 69.7 |
| MLP | 82.2 | 73.3 | **83.1** | 71.9 |
| Transformer | 81.2 | 72.4 | 82.7 | 71.5 |
| Two linear | **83.7** | **74.8** | 82.8 | **72.1** |

(c) Generation of the OOD semantic embeddings.

| Methods | UCF | HMDB | K600 |
|---|---|---|---|
| LLM | 83.7 | 53.2 | 74.6 |
| WebVid-10M | 84.2 | 53.8 | 75.5 |
| Random | 75.2 | 46.8 | 69.2 |
| $\mathcal{N}(0, I)$ | **84.8** | **54.2** | **75.6** |

(d) Effects of regularizing the OOD space in distillation.

| Type | UCF | HMDB | K600 |
|---|---|---|---|
| L2 Loss | 83.6 | 52.8 | 75.0 |
| KL divergence | 83.3 | 53.1 | 74.9 |
| $\mathcal{L}_{\text{RSD}}$ | **84.8** | **54.2** | **75.6** |

(e) Effects of the OOD embeddings number $N$.

| Number | UCF | HMDB | K600 |
|---|---|---|---|
| 100 | **84.8** | 53.9 | 75.4 |
| 200 | **84.8** | **54.2** | 75.6 |
| 400 | 84.6 | 54.0 | **75.7** |

(f) Effects of using rephrased text and weight ensemble.

| Methods | UCF | HMDB | K600 |
|---|---|---|---|
| None | 84.8 | 54.2 | 75.6 |
| +Rephrased text | **86.0** | 54.3 | 77.4 |
| +Weight ensemble | 85.9 | **54.6** | **78.1** |

## 4.3 ABLATION STUDIES

**Applicability to various adaptation methods.** To demonstrate the scalability of our method, we integrate *TACO* with representative parameter-efficient adaptation techniques, including adapter-based, prompt-based, and LoRA-based approaches. As presented in Table 4, the concise design of our method enables a direct integration with various methods and yields consistent performance improvements. Besides, we observe that our method achieves the optimal results when combined with a fully fine-tuned approach. This suggests that when transferring CLIP to the video domain, the model requires sufficient capacity to strike a balance between fitting the video bias and keeping the pre-trained generalization.

**Component-wise analysis of *TACO*.** To analyze the effect of each proposed component, we perform in-depth ablations with the ViT-B/16 network in Table 5a. Our baseline is the ViT-B/16 CLIP model adapted under the standard fine-tuning paradigm. The results show that the Specialization Projection and Relative Structure Distillation contribute respectively to the generalization performance in terms of introducing new knowledge and preserving generalization. Their combination yields a promising synergistic effect, demonstrating the effectiveness of our method.

**Various implementations of the specialization projection.** We ablate the different types of specialization projection in Table 5b. Two baseline models are adopted in this study, differing in whether the text encoder is fine-tuned. When the text encoder is frozen during the fine-tuning process, all three implementations can lead to performance improvements. Instead, MLP and Transformer lead to degraded generalization when the text encoder is tunable. We believe this is due to the activation function leading to a substantial shift to the projected embeddings, which in turn distorts the alignment between text and pre-projected video representations. Moreover, two simple linear layers can effectively decouple the representation and optimization spaces, delivering consistent improvements over both baselines.

**Various implementations of the specialization projection.** We study the effects of different OOD embedding generation methods in Table 5c. We first generate about 3000 real video categories by prompting the LLM model (Hurst et al., 2024) and collect the text descriptions in a large-scale video-text dataset (Bain et al., 2021), and then utilize the K-means (McQueen, 1967) clustering to reduce the number of the embeddings. The results show that our proposed Gaussian sampling method outperforms the OOD categories with authentic semantics, and does not require additional computation. Besides, we try to replace the $z^\top z$ in Equation 5 with a randomly initialized tensor of the shape $D \times D$. The model collapsed during training, indicating that OOD embeddings must reside in the same semantic space as the fine-tuned model.

**Effects of regularizing the OOD space in distillation.** In Table 5d, we show that keeping the space structure of the OOD space in distillation is crucial for preserving generalization. Compared to the vanilla knowledge distillation techniques using the L2 or KL divergence objective, our method can effectively curb the deviation in the OOD space and achieve better performance.

**Varying the number of the OOD samples.** We ablate the number of OOD samples $N$ in Table 5e. $N = 200$ is sufficient to represent the structure of the sampled OOD space and yields the best result.

**Effects of the rephrased text and weight ensemble.** To further enhance the generalization performance, we incorporate the rephrased text descriptions from FROSTER (Huang et al., 2024) and the weight ensemble technique (Weng et al., 2023) into our method. Each can provide additional performance improvements. Besides, for the weight ensemble, we found it to be quite effective when the model is over-trained. But when the model does not exhibit obvious overfitting, it may impose a negative effect. In contrast, our method can be applied to models under any condition and leads to the generalization improvement.

## 5    RELATED WORK

**Adapting VLMs to video recognition.** Adapting VLMs to video recognition tasks has been shown to be effective. In this paradigm, the main challenges lie in effectively injecting the video-specific knowledge and preserving the original generalization inherent in VLMs (Rasheed et al., 2023). For the former, a line of research models the video temporal dynamics by incorporating additional parameterized modules, such as well-designed adapters (Pan et al., 2022; Yang et al., 2022), prompts (Ju et al., 2022a; Wasim et al., 2023; Ju et al., 2022b), and the Transformer layers (Wu et al., 2023a;b; Zhu et al., 2024). For example, X-CLIP Ni et al. (2022) proposes an attention-based and multi-frame integration module for cross-frame information exchange. For the latter, Open-VCLIP (Weng et al., 2023) seeks a middle ground between generalization and specialization by interpolating the model weights along its optimization trajectory. FROSTER Huang et al. (2024) alleviates the overfitting by ensuring the learned features do not diverge too far from the frozen CLIP through knowledge distillation. Open-MeDe Yu et al. (2025) leverages the meta-optimization to mitigate the inherent static bias of the pre-trained model during adaptation. Despite achieving remarkable results in open-vocabulary evaluation, we believe their adaptation effects remain constrained by the inconsistency between the fine-tuning and evaluation objectives. Our method offers new insights into the above challenges and delivers significant improvements in generalization capability.

**Knowledge distillation for VLMs.** Applying knowledge distillation constraints for adapting pre-trained models has been widely explored (Pei et al., 2023; Li et al., 2024), with the aim of enhancing the generalization capability. The core principle is to have the student model mimic the teacher's logits or features, thereby transferring the teacher's generalized knowledge to the student (Yang et al., 2024; Mistretta et al., 2024; Dai et al., 2022) or regularizing the student to not deviate from the teacher (Huang et al., 2024; Addepalli et al., 2024). For example, CLIPPING (Pei et al., 2023) transfers the plentiful knowledge from a larger model to a computationally efficient student model (MobileViT) through layer-wise alignment. FROSTER (Huang et al., 2024) prevents the model from overfitting in fine-tuning by designing the residual feature distillation. However, these methods typically focus on aligning features or logits within the training data distribution. Instead, we emphasize the importance of the OOD space for preserving the generalization, and propose a novel distillation objective to maintain the relative structure of the OOD space during adaptation.

## 6    CONCLUSION

In this work, we present *TACO*, a simple yet effective framework designed to address the inconsistency between fine-tuning and evaluation objectives. By analyzing the limitations of existing paradigms, we formulate a concrete adaptation principle and introduce a task decoupling strategy with a specialization projection to alleviate knowledge overfitting. Furthermore, we propose *Relative Structure Distillation*, which preserves generalization by maintaining consistent relative structures between ID and OOD embedding spaces. Extensive experiments demonstrate that *TACO* achieves state-of-the-art performance across diverse benchmarks under cross-dataset and base-to-novel settings.

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

## A  LLM USAGE STATEMENT

In this work, large language models (LLMs) were used solely as general-purpose tools for language polishing and improving the clarity of writing. All other uses of LLMs, if any, have been explicitly stated in the main text. The authors take full responsibility for the content of this paper.

## B  LIMITATION AND BROADER IMPACT

**Limitation**   While our approach achieves strong performance on open-vocabulary tasks, we observe that its improvements are relatively limited on datasets where temporal information plays a critical role. A promising direction for future research is to integrate our framework with existing temporal modeling techniques to better handle such scenarios. In addition, our method requires setting a relatively small momentum parameter for the teacher model to maintain consistency in the OOD space. However, we observe that the optimal value of this parameter may vary across datasets, which could limit the robustness of our approach. A potential future direction is to develop a more stable strategy for updating the teacher model.

**Broader Impact**   Adapting foundation models to downstream tasks has become a prevailing trend in machine learning. We argue that exploring effective strategies for adapting vision-language models to open-vocabulary tasks is both timely and necessary for real-world applications. This work seeks to offer insights that contribute to the broader and long-term use of foundation models. While our study focuses on video recognition, which has wide-ranging applications such as surveillance, it is crucial that concerns regarding privacy and individual rights are thoroughly considered prior to practical deployment.

Table 6: Hyper-parameter details during fine-tuning.

|  | Value |
|---|---|
| *Optimization details* | |
| Batch size | 192 |
| Optimizer | AdamW |
| Weight decay | 0.2 |
| Adam $\beta_1, \beta_2$ | 0.9, 0.999 |
| Learning rate (Projection) | 5e-5 |
| Learning rate (CLIP layers) | 3.75e-6 |
| Learning rate decay | Cosine |
| Training epochs | 15 |
| Linear warm-up epochs | 5 (cross-dataset), 2 (bas-to-novel) |
| *Augmentation* | |
| RandomResizedCrop | |
|     Area | [0.08, 1.00] |
|     Aspect ratio | [3/4 , 4/3] |
|     Crop size | 224 |
| Random Horizontal Flip | 0.5 |
| Random Gray scale | 0.2 |

## C  MORE IMPLEMENTATION DETAILS

In Table 6, we present the hyper-parameters set for optimization. For both the Cross-dataset and Base-to-novel settings, we trained the model for 15 epochs. All experiments are conducted using 3 NVIDIA GeForce RTX 4090.

For cross-dataset evaluation, the methods are evaluated on three official splits or the full dataset of UCF-101 and HMDB-51. For Kinetics-600, we adopt the three splits provided by (Chen & Huang, 2021). Each split contains 160 categories out of 220 new categories that do not exist in K400. We report the average Top-1 accuracy and the standard deviation on three splits. To further enhance the

generalization performance, we incorporate the rephrased text descriptions from FROSTER (Huang et al., 2024) and the weight ensemble technique (Weng et al., 2023) into our method.

For base-to-novel evaluation, we do not apply weight ensemble since it may affect the model's performance on the base category. Following the previous work (Huang et al., 2024), we employed the rephrased text descriptions in this setting. Besides, we do not apply the rephrased text descriptions to the SSv2 dataset.

## D  ADDITIONAL ABLATIONS

**Training cost analysis of TACO.**  We report the actual training time of our method with respect to the baseline in Table 7. The wall-clock time of training is benchmarked on 3 4090 GPUs with a batch size of 192. GPU days are calculated by the number of GPUs multiplied by the training time in hours. As shown in the table, applying the specialization projection does not introduce additional training overhead over the baseline, due its light implementation. Incorporating $\mathcal{L}_{\text{RSD}}$ brings a +1.4 days training time increase since it requires an additional forward pass for the teacher model.

Table 7: Ablation study on the training costs of TACO.

| Method | GPU-hours |
|---|---|
| Finetuned CLIP | 36.3 |
| + Specialization Projection | 36.4 |
| + $\mathcal{L}_{\text{RSD}}$ | 37.8 |

**Effects of varying momentum coefficients.**  In this study, we explored the impact of different teacher models on generalization performance and the effects of the momentum parameter. As shown in the table, using the frozen model as the teacher model (i.e. momentum=1.0) yields inferior results. We believe this is because its supervision signals constrain the model's ability to learn new knowledge. Using the EMA update strategy strikes a good balance between maintaining consistency and incorporating new knowledge. Besides, we found that using a smaller momentum parameter yields better results, indicating the importance of preserving the consistency of the OOD space structure.

Table 8: Effects of varying momentum coefficients.

| Momentum | UCF | HMDB |
|---|---|---|
| 1.0 | 83.8 | 53.1 |
| 0.99998 | 84.2 | 54.0 |
| 0.9998 | 84.8 | 54.2 |
| 0.998 | 84.1 | 53.8 |

## E  TEXTUAL PROMPTS USED IN EVALUATION

Following the previous work (Zhu et al., 2024), we adopt a set of hand-craft textual prompt templates to generate text embeddings during the evaluations. Following CLIP (Radford et al., 2021), we perform prompt ensembling over the 28 templates in order to provide comprehensive semantics. The templates are listed in Table 9.

## F  DATASET DETAILS

**Kinetics-400**  (Kay et al., 2017) is a large-scale dataset in the video domain. The dataset contains ∼240k training videos and ∼20k validation videos in 400 human action categories, with an average length of 10 seconds. The high quality of the dataset makes it the most popular benchmark for video recognition

**Kinetics-600**  (Carreira et al., 2018) is an extension of Kinetics-400, consisting of ∼392k training videos, ∼30k validation videos, and ∼60k test videos in 600 human action categories. The dataset contains an additional 220 new action categories over Kinetics-400. We evaluate the zero-shot performance on 220 new categories and adopt three splits provided by the previous work (Chen & Huang, 2021). We use its test set for evaluation and report the average performance on three splits.

**UCF-101**  (Soomro et al., 2012) is an action recognition dataset that contains 13,320 videos in 101 action categories, collected from YouTube. There are three official splits of training data and validation data.

Table 9: Textual prompt templates of TACO.

| Templates |
| --- |
| 'a photo of {category}.' |
| 'a photo of a person {category}.' |
| 'a photo of a person using {category}.' |
| 'a photo of a person doing {category}.' |
| 'a photo of a person during {category}.' |
| 'a photo of a person performing {category}.' |
| 'a photo of a person practicing {category}.' |
| 'a video of {category}.' |
| 'a video of a person {category}.' |
| 'a video of a person using {category}.' |
| 'a video of a person doing {category}.' |
| 'a video of a person during {category}.' |
| 'a video of a person performing {category}.' |
| 'a video of a person practicing {category}.' |
| 'a example of {category}.' |
| 'a example of a person {category}.' |
| 'a example of a person using {category}.' |
| 'a example of a person doing {category}.' |
| 'a example of a person during {category}.' |
| 'a example of a person performing {category}.' |
| 'a example of a person practicing {category}.' |
| 'a demonstration of {category}.' |
| 'a demonstration of a person {category}.' |
| 'a demonstration of a person using {category}.' |
| 'a demonstration of a person doing {category}.' |
| 'a demonstration of a person during {category}.' |
| 'a demonstration of a person performing {category}.' |
| 'a demonstration of a person practicing {category}.' |

**HMDB-51** (Kuehne et al., 2011) contains 7k videos in 51 action categories, collected from movie clips and web videos. There are three official splits of the dataset, each with 3,570 training data and 1,530 validation data. is a collection of realistic videos from various sources, including movies and web videos. The dataset comprises 7,000 video clips from 51 action categories.

**Somethin-Something V2** (Goyal et al., 2017) is a temporal-heavy dataset that requires the fine-grained temporal understanding capability of the model. It contains 220,000 videos in 174 action categories.

