# OpenReview forum: "Towards Task-Consistent Open-Vocabulary Adaptation in Video Recognition"
_ICLR.cc/2026/Conference — ICLR 2026 Conference Withdrawn Submission_

### Official Review · Reviewer_Tfit · 2025-10-31

**Soundness:** 3
**Presentation:** 3
**Contribution:** 3
**Rating:** 6
**Confidence:** 4

**Summary:**

Transferring CLIP for open-vocabulary video recognition has shown strong effectiveness, but its standard fine-tuning paradigm on video datasets overlooks representation space variations beyond the training distribution, leading to sub-optimal adaptation. Existing fine-tuning methods fail to address the inconsistency between fine-tuning and evaluation objectives, which causes potential negative effects and prevents them from meeting the generalization needs for unseen categories. To mitigate this issue, the proposed TACO framework adopts a clear adaptation principle: it uses a task decoupling method (with a specialization projection) to reduce knowledge overfitting, and introduces Relative Structure Distillation to maintain consistent relative structures between in-distribution and out-of-distribution representation spaces via knowledge distillation. Experiments demonstrate that TACO achieves state-of-the-art performance on various benchmarks under both cross-dataset and base-to-novel settings, verifying its effectiveness for CLIP-based open-vocabulary video recognition.

**Strengths:**

1. This paper is well written, well organized and easy to follow. The motivation is clear.

2. It is interesting that the authors reveal that “ replacing the fine-tuned text encoder with  CLIP’s text encoder has almost no impact on the results  (66.3%vs. 66.2%),which suggests that the OOD generalization of the adapted model is grounded in CLIP’s original semantic space”.

3. The obseration of Figure 4 is valuable.

4. The proposed method is simple but effective and acheive good results. The experimental results are extensive.

**Weaknesses:**

1. Eqn. 1 v=f_{theta}_v (V) is not appropriate, since f_{theta}_v  is an image encoder and V is a set of images, so v should be an aggretation of f_{theta}_v  (V) like averaging.

2. The proposed method is a general method that is not specific for video. I mean is they any specific design for addressing the temporal model in video. It seems the proposed method could directly apply to open vocabulary image recognition. Is it possible to evaluate in open vocabulary image recognition benchmark?

3. What is the relationship among Corresponding, Corresponding + K400 (ID) and  ALL in Fig. 2. The authors could provide a example to illustrate these set.

**Questions:**

Please refer to the Weakness.

---

### Official Review · Reviewer_5LBp · 2025-11-03

**Soundness:** 2
**Presentation:** 2
**Contribution:** 2
**Rating:** 2
**Confidence:** 4

**Summary:**

This paper mainly focuses on transferring CLIP for open-vocabulary video recognition, where the model is fine-tuned to achieve better generalization performance on unseen data. However, such a fine-tuning paradigm overlooks the variations of the representation space beyond the distribution of training data. To solve this problem, this paper proposes TACO to mitigate the potential negative effects induced by the inconsistency between the fine-tuning and evaluation objectives.

**Strengths:**

- The paper focuses on achieving a balance between preserving the prior knowledge in the pre-trained CLIP and obtaining more knowledge from the data of open-world tasks, which is an important problem in foundation model fine-tuning.
- The paper is generally well written with clear logic and convincing demonstrations.
- The empirical results look good.

**Weaknesses:**

- The illustration in Fig. 1 is not clear enough. Specifically, I cannot figure out the insights of the paper from the current illustration figure.. The relationship here is not clear enough. The figure can be further polished.
- The notations in the paper are not well managed. For example, the set notation $\mathcal{S}$ is not defined before it is used.
- The motivation can be further enhanced by providing more empirical or theoretical results. Current studies, such as "in-distribution bias", are the common sense in machine / deep learning communities.
- The analysis studies in Section 2.2 are a little bit confusing.
- The proposed TACO is not convincing enough due to a lack of insights. Meanwhile, the proposed method is not novel.

**Minor Problems:**
- In the paper, "fine-tune" should be written as "fine-tuning". For example, "the fine-tune paradigm" should be "the fine-tuning paradigm"
- Lack definition of $one hot$ function in Eq. (2).  Although everyone knows it, it would be better to provide comprehensive definitions in the research paper

**Questions:**

- A problem in open-vocabulary video recognition with CLIP is introducing video-specific knowledge while preserving the original pre-trained knowledge. In other words, the root problem here is fine-tuning original models may result in forgetting, and the ultimate goal here is achieving a trade-off between learning more new knowledge and preserving as much knowledge from pre-training as possible.
- A following question here is why you define the forgetting problem of foundation model fine-tuning as an inconsistency problem?
- Compared to the common sense that models cannot achieve satisfactory performance on data from distributions that are different from the training data, what are the new insights of your first analysis study in Section 2.2?
- For the second study in Section 2.2, the finding that replacing the fine-tuned text encoder with the original text encoder does not affect the performance is interesting. However, such a phenomenon is not consistent in the case of the original visual encoder with the fine-tuned/original visual encoder. Thus, does this indicate the drawbacks of the paradigm that jointly fine-tunes both encoders?
- How do you achieve the hypothesis in the third study of Section 2.2?
- In the third study of Section 2.2, how do you measure the similarity between models? Why does the phenomenon motivate the summary in this part?
- As mentioned, the proposed specialization projection is inspired by contrastive learning. Thus, what are the relations between the problems studies in this paper and the insight of adopting the projection?
- What are the OOD semantic embeddings? How do you sample OOD semantic embeddings from the training data distribution?

---

### Official Review · Reviewer_raHh · 2025-11-06

**Soundness:** 3
**Presentation:** 2
**Contribution:** 2
**Rating:** 4
**Confidence:** 3

**Summary:**

This paper proposes TACO (Task-Consistent Open-Vocabulary Video Adaptation), which aims to mitigate the inconsistency between fine-tuning and evaluation objectives when transferring CLIP models to open-vocabulary video recognition. TACO introduces two components:
1. Specialization Projection, a linear projection head that decouples the representation and optimization spaces to reduce overfitting.
2. Relative Structure Distillation (RSD), which maintains consistent relative alignment between in-distribution (ID) and out-of-distribution (OOD) embeddings via knowledge distillation.

Experiments show that TACO achieves state-of-the-art performance on several cross-dataset and base-to-novel benchmarks.

**Strengths:**

1. The method consistently improves baselines across multiple datasets.
2. The empirical diagnosis on the text encoder contribution appears to provide useful observations for understanding VLM adaptation in videos.

**Weaknesses:**

1. In my view, the claimed “inconsistency between fine-tuning and evaluation objectives” essentially restates the known overfitting/generalization trade-off. Prior works such as FROSTER already aim to prevent overfitting by regularizing fine-tuning with knowledge distillation. Therefore, it seems doubtful to say the issue has been overlooked in previous papers.
2. The "OOD semantic embedding generation" is somewhat underspecified. It samples random unit vectors from a Gaussian distribution and calls them “OOD embeddings,” but these vectors have no semantic meaning. To my understanding, this step essentially serves as a geometric regularizer rather than truly modeling out-of-distribution semantics.
3. Missing ablation: (a) The method discards the projection head during evaluation. Representation comparisons before vs. after the projection might be needed to verify the generalization effect. (b) An ablation on the EMA teacher is needed: EMA teacher vs. frozen teacher.

Minors:

1. Figure 1 is not referenced; it is unclear what point it is used for.
2. Figure 3: “Text sim: 1” seems to be a typo?

**Questions:**

1. Could the authors clarify whether the “inconsistency” issue is fundamentally different from the overfitting/generalization trade-off discussed in previous open-vocabulary video adaptation works?
2. How do the random hypersphere-sampled OOD embeddings relate to real unseen categories in CLIP’s text space?

---

### Note · Authors · 2025-11-14

I have read and agree with the venue's withdrawal policy on behalf of myself and my co-authors.